# Evaluation of the Utilization of Near-Infrared Fluorescent Contrast Agent ASP5354 for In Vivo Ureteral Identification in Renal Diseases Using Rat Models of Gentamicin-Induced Acute Kidney Injury

**DOI:** 10.3390/diagnostics13101823

**Published:** 2023-05-22

**Authors:** Katsunori Teranishi

**Affiliations:** Graduate School of Bioresources, Mie University, 1577 Kurimamachiya, Tsu 514-8507, Japan; teranisi@bio.mie-u.ac.jp; Tel.: +81-59-231-9615

**Keywords:** ASP5354, diagnosis, kidney injury, near-infrared fluorescence, real-time imaging, ureteral identification

## Abstract

ASP5354 was recently developed as a near-infrared fluorescence (NIRF) contrast agent for intraoperative ureteral identification, and its use has been evaluated in healthy animals. However, the utilization of ASP5354 for ureteral identification has not been evaluated in animals with renal injury. In this study, we assessed the application of ASP5354 for ureteral imaging using rat models of gentamicin-induced mild, moderate, and severe acute kidney injury (AKI), using a clinically available NIRF detection system. NIRF was detected in the abdominal cavity and ureters after laparotomy, and the efficiency of ASP5354 was evaluated based on the NIRF signal intensity over 60 min. After the intravenous injection of ASP5354 into rats with mild or moderate AKI, the ureters were clearly imaged at a high ratio of NIRF intensity in the ureter to that in the tissues around the ureter. Six days after intravenous injection, the use of ASP5354 in rats with moderate AKI did not affect the biochemical kidney functions or histopathological conditions of the kidney tissues, as compared to those with no injection of ASP5354. In rats with severe AKI, ureteral imaging was not effective due to the relatively strong NIRF expression in the tissues around the ureters. These data indicate that ASP5354 holds potential as a contrast agent for intraoperative ureteral identification in patients with limited renal injury.

## 1. Introduction

The incidence of intraoperative iatrogenic ureteral injuries (IUIs) during urologic, colorectal, or gynecologic surgery may be >1% [1,2]. Additionally, the risk of IUI occurrence is significantly higher during laparoscopic abdominal surgery than that during open surgery [3]. A report on ureteral injuries occurring during abdominal surgery indicated that 62% of injuries were identified postoperatively [4], making them more difficult to treat than injuries detected early. Therefore, early intraoperative ureteral diagnosis and identification are necessary for ureteral repair without complications.

Intraoperative ureteral identification and diagnosis are required in order to prevent IUIs and to detect renal injuries early. Ureteral stent placement has previously been employed; however, this technique results in a longer operating time, a higher risk of acute kidney injury (AKI), and urinary tract complications, as well as higher costs [5,6,7]. The ureters were intraoperatively identified based on the fluorescence of methylene blue (MB), a diagnostic agent approved by the Food and Drug Administration, present in urine after the intravenous administration of MB [8,9,10,11]. Because MB emits fluorescence at a *λ*_max_ of 690 nm, which easily interferes with tissues and is selectively excreted by the liver, it is unsuitable for efficient intraoperative ureteral identification and diagnosis [1].

Significant efforts have been made toward urological disease monitoring, including intraoperative ureteral identification using near-infrared fluorescence (NIRF, 700–900 nm) [12,13,14,15,16,17,18,19,20,21,22,23,24,25,26,27,28], which demonstrates deep tissue penetration, low autofluorescence in tissue, and low tissue scattering [29]. Near-infrared fluorescent CW800 CA and UreterGlow together facilitate intraoperative ureteral identification in rats and pigs [17,18,19,20,21]. Following intravenous injection, these fluorescent probes compete for clearance by the liver. Near-infrared fluorescent ZW800-1 [22], UL-766 [23], and ZW800-1C [24] are selectively cleared from the kidneys of rats and mice after intravenous injection. In 2019, a clinical study utilizing ZW800-1 for intraoperative ureteral identification in humans was reported [25]. In contrast, ZW800-1C demonstrated intraoperative ureteral diagnosis by intravenous injection in mouse models of physical ureteral injury. However, the tissues surrounding the ureters emitted nonspecific fluorescence, resulting in a low contrast-to-background ratio [26].

We developed a near-infrared (NIR) fluorescent contrast agent, CD-NIR-1, for intraoperative ureteral identification [30]. In the molecular structure of CD-NIR-1, hydrophobic heptamethine indocyanine moiety is coated with hydrophilic *β*-cyclodextrin moiety, which can include hydrophobic molecules [31]. Due to this structural property, CD-NIR-1 is rapidly excreted via the kidneys and ureters into the bladder following intravenous injection, thus emitting strong NIRF in the ureters. Subsequently, ASP5354, formerly designated as TK-1 [32], which has structural and renal excretion properties similar to CD-NIR-1, was developed, and assessed in animal and phase 1 studies [33,34]; currently, a phase 3 study is underway [35].

Although ASP5354 demonstrated a significant capacity for intraoperative ureteral identification in normal animals [33], the limitations of intraoperative ureteral identification in animals with renal dysfunction remain unclear. Aminoglycoside antibiotics, including gentamicin (GM), are widely used as antibacterial agents for treating gram-negative infections [36]. These agents increase the risk of nephrotoxicity; in some patients, they may trigger AKI, defined as a sudden reduction in renal function or glomerular filtration rate [37,38]. In this study, we evaluated the use of ASP5354 for intraoperative ureteral identification in a rat model of GM-induced AKI (Figure 1).

## 2. Materials and Methods

### 2.1. Experimental Animals

This study complied with the regulations on animal experimentation of the Mie University Animal Ethics Committee (registration no. 22–63). Male Wistar rats (age, four weeks; mean weight, 80 g) were obtained from Japan SLC, Inc. (Shizuoka, Japan). The rats were housed at 21–23 °C with free access to food and water. Before the experiments were performed, the rats were anesthetized by subcutaneous injection of pentobarbital sodium salt dissolved in saline.

### 2.2. Materials

ASP5354 was prepared as described previously [32]. It was rapidly dissolved in saline at 24 μmol/L and used immediately. GM (sulfate compound) was purchased from Wako Chemicals Co., Ltd. (Osaka, Japan), and dissolved at 0.05 g/mL in saline. The food (MF) for the rats was purchased from Oriental Yeast Co., Ltd. (Tokyo, Japan).

### 2.3. Instruments

The NIRF images were obtained using a clinically available Photodynamic Eye camera system (Hamamatsu Photonics K.K., Shizuoka, Japan) in a dark box. The distance between the camera and the experimental animals was set at 10 cm. The video images were recorded on a personal computer. The measurement conditions were as follows: brightness, 5 a.u.; contrast, 5 a.u.; and excitation level, 1.0–4.0 a.u. The NIRF intensity was analyzed using a region-of-interest analysis program (Hamamatsu Photonics K.K.). Photomicroscopy was performed using an OLYMPUS Bx61VS microscope (Tokyo, Japan). In order to analyze the amount of ASP5354 in urine, high-performance liquid chromatography (HPLC) was performed at 25 °C on an EXTREMA HPLC system equipped with an FP-1520 fluorescence detector (Jasco, Tokyo, Japan), controlled by a ChromNAV Ver.2 system (Jasco). A Cosmosil 5C18-MSII column (4.6 × 150 mm; Nacalai Tesque, Inc., Kyoto, Japan) was used for separation.

### 2.4. Generation of Rats with AKI and Analysis

GM (0.05 g/mL in saline) was intraperitoneally administered at single doses (0.25 g/kg body weight per dosage) at 24 h intervals for 5, 7, or 8 total doses. In the control group, saline was injected instead of GM. In order to determine the degree of AKI, the mean weight of the left and right kidneys (each group, *n* = 5 rats), levels of blood urea nitrogen (BUN) (each group, *n* = 5 rats), creatinine in serum (each group, *n* = 5 rats), and creatinine in urine (each group, *n* = 5 rats), were measured, and visual histological analysis of the kidneys was performed 24 h after each final administration of GM. BUN levels were measured using the QuantiChrom^TM^ Urea Assay Kit DIUR-100 (BioAssay Systems, Hayward, CA, USA) according to the improved Jung method, which utilizes a chromogenic reagent to form a colored complex specifically with urea. The color intensity was measured at 520 nm and was found to be directly proportional to the urea concentration in the blood. Serum and urinary creatinine levels were measured using creatinine (serum) colorimetric assay kits (Cayman Chemical Company, Ann Arbor, MI, USA) and creatinine (urinary) colorimetric assay kits (Cayman Chemical Company), respectively. These assays rely on the Jaffe reaction, in which a yellow/orange color is formed when the metabolite is treated with alkaline picrate. The color of the dye was measured at 490 nm. For histological analysis, kidneys (each group *n* = 3 rats) were removed, fixed in 10% neutral formalin, and processed. Paraffin sections 3 µm thick were stained with hematoxylin and eosin or periodic acid-Schiff reagent. Histological analyses were performed at the Applied Medical Research Laboratory (Osaka, Japan) by visual inspection.

### 2.5. Assessing the Amount of ASP5354 in Urine

Urine was collected 0–3, 3–6, and 6–9 h after intravenous injection of ASP5354 at a single dose of 120 nmol/kg body weight 24 h after GM administration at single doses of 0.25 g/kg body weight at 24 h intervals 5, 7, or 8 times (each group *n* = 5 rats). The amount of ASP5354 in urine was analyzed using an HPLC system. Elution was performed at 0.8 mL/min using 0.1% trifluoroacetic acid (TFA) in water (solvent A) and 0.1% TFA in acetonitrile (solvent B). The elution for analytical HPLC began with 80% A, followed by a linear gradient to 100% B over 30 min. The excitation and emission wavelengths for fluorescence analysis were 800 and 820 nm, respectively.

### 2.6. In Vivo NIRF Ureter Imaging

In a previous report, CD-NIR-1 was intravenously injected at a single dose of 25 nmol/kg body weight into normal rat tails for in vivo NIRF imaging of ureters [30]. In the present study, owing to using ASP5354, which has slightly longer absorption and emission wavelengths than that of CD-NIR-1, on rats with AKI, ASP5354 was intravenously injected at a single dose of 40 or 120 nmol/kg body weight into the rat tail for the ureter imaging experiments. NIRF images of the abdominal cavity, centered on the ureters, were obtained using a Photodynamic Eye camera system (each experiment: *n* = 3 rats) 5, 10, 20, 30, 40, and 60 min after laparotomy. NIRF intensities in the kidneys, ureters, and tissues surrounding the ureters were analyzed.

NIRF images of the abdominal cavity were obtained using a Photodynamic Eye camera system after laparotomy (each experiment: *n* = 3 rats) 24 h after the intravenous injection of ASP5354 at a single dose of 120 nmol/kg body weight in AKI rats. NIRF intensities in the kidneys were analyzed.

### 2.7. Assessing Effects of ASP5354 on AKI

Six days after intravenous injection of saline (5 mL/kg body weight) or ASP5354 at a single dose of 120 nmol/kg body weight 24 h after 7 intraperitoneal administrations of GM at single doses of 0.25 g/kg body weight at 24 h intervals, kidney weight, BUN, serum creatinine, and urinary creatinine levels were measured. The kidneys were removed and histological analysis was performed.

### 2.8. Statistical Analysis

Data on kidney weight, concentrations of BUN, serum creatinine, urinary creatinine, ASP5354 in urine, and NIRF intensity in vivo are presented as mean values ± standard deviation (SD). Statistical analysis was performed using the Student’s *t*-test. *p*-values were considered statistically significant.

## 3. Results

### 3.1. Kidney Characteristics of Rat Model with GM-Induced AKI

Rats with three stages of AKI, including mild, moderate, and severe, were generated by varying the number of GM administrations at single doses of 0.25 g/kg body weight at 24 h intervals, i.e., 5, 7, and 8 doses, respectively. Kidney weight (Figure 2a), BUN level (Figure 2b), and serum creatinine level (Figure 2c) increased significantly with the severity of AKI compared to those in the control rats, and the level of urinary creatinine decreased with the severity of AKI compared to that in the control rats (Figure 2d). Compared to the control rats (Figure 3a–c and Figure 4), proximal tubular denaturation, such as hypertrophy of the proximal tubular epithelium cells, resulting in a reduction in tubular cavity, was observed in many areas of the kidneys of rats with mild AKI (Figure 3d–f and Figure 4), and tubular necrosis was observed in the kidneys of rats with moderate AKI (Figure 3g–i and Figure 4). In the kidneys of rats with severe AKI, tubular epithelial cell hypertrophy and necrosis were more frequent (Figure 3j–l and Figure 4). Glomerular degeneration, evidenced by the hypercellularity of endothelial cells in the glomerular capillaries, was observed in all three stages. The excretion rate of ASP5354 from the body as a urinary component was significantly reduced with the severity of AKI compared to that in the control rats (Figure 5); however, there was no significant difference in the total amount of ASP5354 excreted 0–9 h after ASP5354 injection (Figure 5).

### 3.2. In Vivo NIRF Imaging of Ureters in Normal Rats

After ASP5354 was intravenously injected into normal rats, ASP5354 was immediately transferred to the kidneys and, subsequently, to the ureters, resulting in NIRF emission of ASP5354 in the kidneys and ureters. The urine flowed intermittently rather than continuously through the ureters (Figure 6). Therefore, in this study, the NIRF intensity in the ureters was analyzed at the midpoint between the kidney and a bladder filled with urine. Because ASP5354 at a single dose of 120 nmol/kg body weight demonstrated strong NIRF emission in the ureters, the fluorescence intensity exceeded the detection limit, even at an excitation of 1 a.u. on the NIRF camera system. When ASP5354 was intravenously injected at a single dose of 40 nmol/kg body weight, NIRF in the ureters was observed above an excitation of 2.0 a.u., and the NIRF intensity in the ureters exceeded the detection limit, even at an excitation of 2.0 a.u. (Figure 7), due to the significant concentration of ASP5354 in the ureters. Therefore, in this study (including all imaging experiments), the quantification of NIRF intensity in the ureters of all rats was difficult under the same measurement conditions, such as the same excitation level on the camera system.

### 3.3. In Vivo Imaging of Ureters in AKI Rats

NIRF imaging of rats with mild and moderate AKI using an intravenous injection of ASP5354 at a single dose of 120 nmol/kg body weight demonstrated that the persistence of ASP5354 NIRF in the ureters was shortened with the severity of AKI, and increased at an excitation of 2.0 a.u., which was required for clear NIRF detection at 60 min (Figure 8a–c and Figure 9a,b). It was difficult to observe a difference in NIRF intensity between the ureters and surrounding tissues in rats with severe AKI, even immediately after ASP5354 injection, due to the strong NIRF emission in the kidneys and surrounding tissues around the ureters (Figure 8d and Figure 9c). As presented in Figure 9b, NIRF intensities in the ureters of rats with severe AKI at 60 min are presented at 100–150 a.u.; however, the intensities may be affected by NIRF emission in the tissues around and/or under the ureters. The NIRF of ASP5354 was not observed in the kidneys of control rats, nor in those with normal, mild, or moderate AKI at an excitation of 3.0 a.u., 24 h after the injection of ASP5354; however, the NIRF intensity in the kidneys of severe AKI rats was significantly high (Figure 10a,b).

### 3.4. Effect of ASP5354 on GM-Induced Nephrotoxicity

Six days after the intravenous injection of ASP5354 24 h after completing seven administrations of GM, the kidney weight, BUN level, serum creatinine level, and urinary creatinine value were unaffected by ASP5354 injection, as compared to saline injection instead of ASP5354 (Figure 11a–d). Histological analysis exhibited the regeneration of proximal tubules 6 days after intravenous injection with either ASP5354 or saline 24 h after the completion of seven administrations of GM, and there was no significant histopathological difference between ASP5354 and saline injections (Figure 12).

## 4. Discussion

Recently, ASP5354 was developed as a NIRF contrast agent for intraoperative ureteral identification using ultrarapid renal excretion properties and assessed in-depth for in vivo ureter imaging in healthy animals and humans [33,34]. Moreover, the potential use of ASP5354 for in vivo cancer imaging has been demonstrated in mouse models [39,40,41]. However, the utility of ASP5354 for in vivo ureter identification has not been evaluated in animals with renal injury. In the present study, we evaluated its efficacy in rat models of GM-induced AKI.

This study demonstrated that intravenous injection of ASP5354 at 120 nmol/kg body weight can clearly identify the ureters in vivo in rats with mild and moderate AKI, as well as in normal rats, but not in those with severe AKI. In this study, mild, moderate, and severe AKI were defined as development caused by 5, 7, and 8 administrations of GM, at single doses (0.25 g/kg body weight) per administration at 24 h intervals, respectively—not defined similarly in human medicine. In mild and moderate AKI, in vivo ureteral identification was possible for at least 60 min after the administration of ASP5354 by properly adjusting the excitation of a clinically available Photodynamic Eye camera system. In rats with severe AKI, relatively strong NIRF in the kidneys and tissues around the ureters interfered with the weak NIRF in the ureters, hampering the identification of the ureters. The strong NIRF in the kidneys and tissues around the ureters indicated that ASP5354 was slowly excreted due to decreased renal function and was retained within the body and kidneys. This explanation was supported by the results obtained in the analysis of the amount of ASP5354 present in urine, i.e., renal excretion of ASP5354 was slower than that in rats with mild and moderate AKI.

ASP5354 of 95, 93, and 94% of injected ASP5354 in rats with mild, moderate, and severe AKI, respectively, was excreted in urine within 9 h after injection of ASP5354. ASP5354 did not affect kidney weight, BUN, serum creatinine, or urinary creatinine levels, or the histopathological conditions of kidney tissues, such as glomerular repair and regeneration of proximal tubules, as observed six days after ASP5354 injection in rats with moderate AKI. These results demonstrated that the administration of ASP5354 is permissible for rats with moderate AKI. However, it is not acceptable in all patients with renal injury. In addition, in this study, we obtained no data indicating whether all ASP5354 was excreted in the urine, which organs contained trace amounts of ASP5354, or whether residual ASP5354 affected the body, due to limitations in analytical techniques. Therefore, the safety and tolerance of ASP5354 need to be evaluated in detail in patients with various types of kidney injuries.

Using rat models of GM-induced AKI, this study demonstrated the ability of ASP5354 to perform in vivo ureteral imaging. However, the appropriate dose of ASP5354 for imaging in humans remains unknown. NIRF imaging experiments were performed using a clinically available NIRF device. However, laparoscopy is generally used in clinical in vivo ureteral imaging. Therefore, in vivo ureteral imaging in clinical laparoscopic settings using NIRF camera systems also needs to be evaluated in patients with renal injuries.

## 5. Conclusions

In conclusion, this study demonstrated the performance of ASP5354 in in vivo ureteral imaging using rat models of GM-induced AKI, indicating its potential for human intraoperative ureteral identification in patients with limited renal injury. Following the intravenous injection of ASP5354 in rats with GM-induced mild and moderate AKI, ureters were clearly imaged using the NIRF of ASP5354 and a NIRF camera system at a high ratio of NIRF intensity in ureters to that in tissues around the ureters. Six days after intravenous injection, ASP5354 in rats with moderate AKI did not affect biochemical kidney functions or the histopathological conditions of kidney tissues, including glomerular repair and regeneration of proximal tubules, as compared to saline injection instead of ASP5354. These data indicate that ASP5354 is a potential contrast agent for intraoperative ureteral identification in patients with limited renal injury.

## Figures and Tables

**Figure 1 diagnostics-13-01823-f001:**
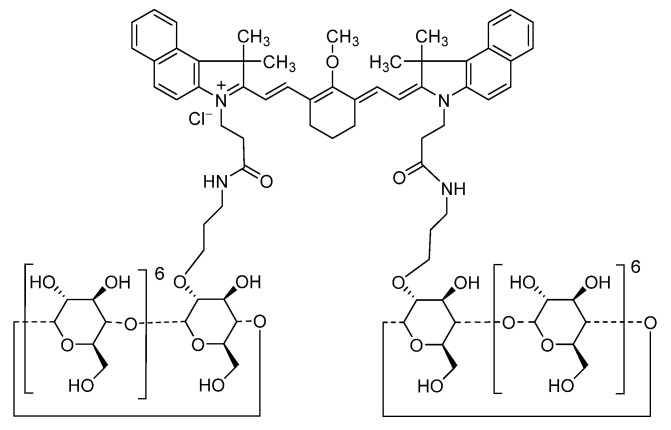
Chemical structure of ASP5354.

**Figure 2 diagnostics-13-01823-f002:**
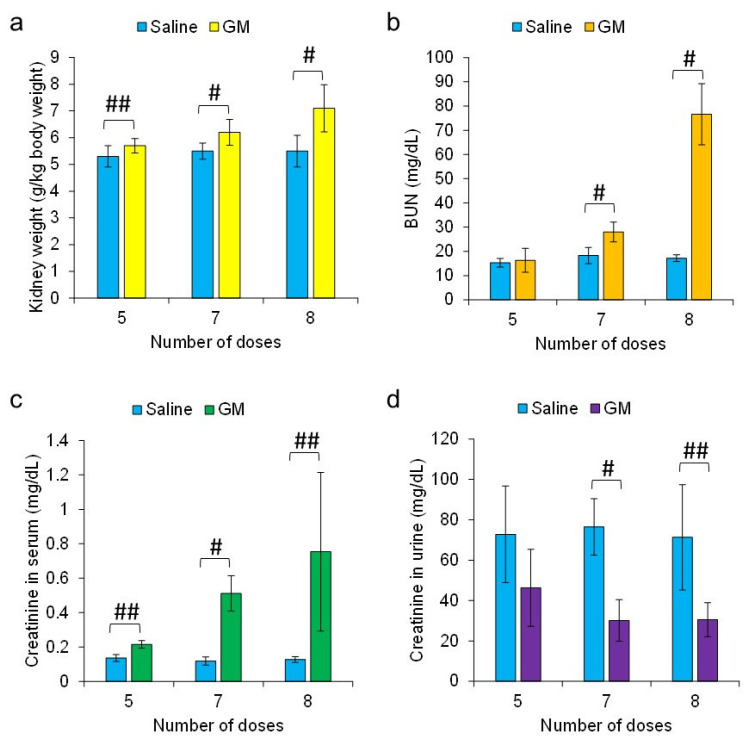
Characteristic parameters of gentamicin (GM)-induced nephrotoxicity. The parameters were measured 24 h after intraperitoneal administration of saline (5 mL/kg body weight per administration at 24 h intervals) or GM at single doses (5, 7, or 8 times) of 0.25 g/kg body weight per administration at 24 h intervals. # *p* < 0.005 and ## *p* < 0.05 indicate significant difference between a pair. (**a**) Kidney weight. (**b**) Blood urea nitrogen (BUN). (**c**) Creatinine in serum. (**d**) Creatinine in urine. Data are expressed as mean ± SD (*n* = 5 rats).

**Figure 3 diagnostics-13-01823-f003:**
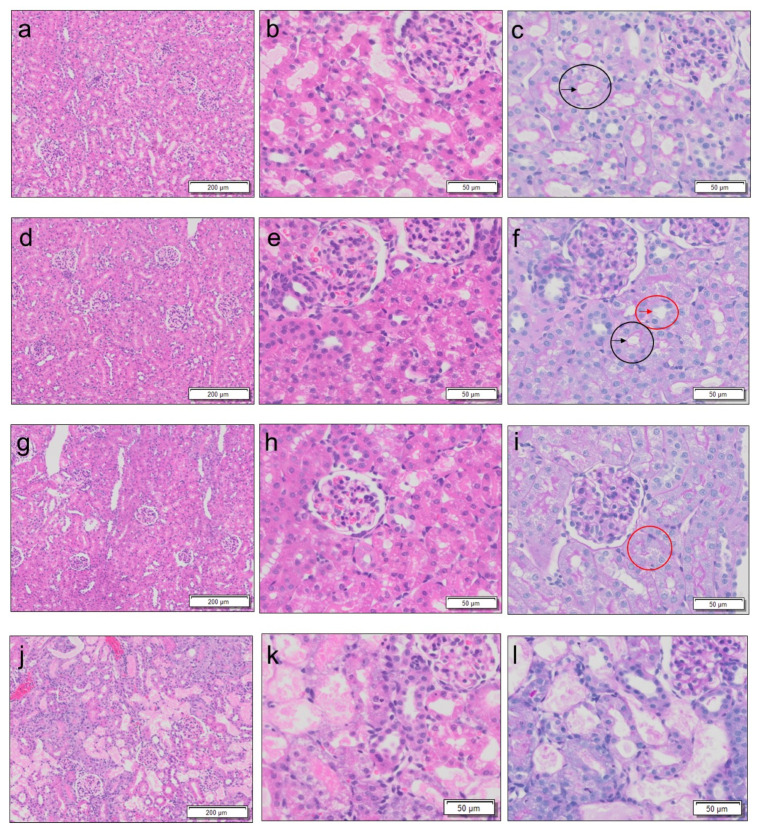
Representative photomicrographs of the kidneys of rats with gentamicin (GM)-induced acute kidney injury. GM was intraperitoneally injected at single doses of 0.25 g/kg body weight per administration at 24 h intervals. (**a**–**c**) 24 h following 5 intraperitoneal injections of saline (0.5 mL/kg body weight per administration at 24 h intervals). Black circle is representative of normal proximal tubules. Black arrow represents the clear brush border of the proximal tubule. (**d**–**f**) 24 h following 5 doses of GM. Black circle indicates atrophy of tubular cavity. Black arrow indicates the brush border of the proximal tubule. Red circle indicates proximal tubular necrosis. Red arrow indicates absence of the brush border of the proximal tubule. (**g**–**i**) 24 h following 7 doses of GM. Red circle represents proximal tubular necrosis. (**j**–**l**) 24 h following 8 doses of GM. There are no normal proximal tubules. The photomicrographs (**a**,**b**,**d**,**e**,**g**,**h**,**j**,**k**) illustrate hematoxylin and eosin-stained tissues. The photomicrographs (**c**,**f**,**i**,**l**) illustrate periodic acid-Schiff-stained tissues.

**Figure 4 diagnostics-13-01823-f004:**
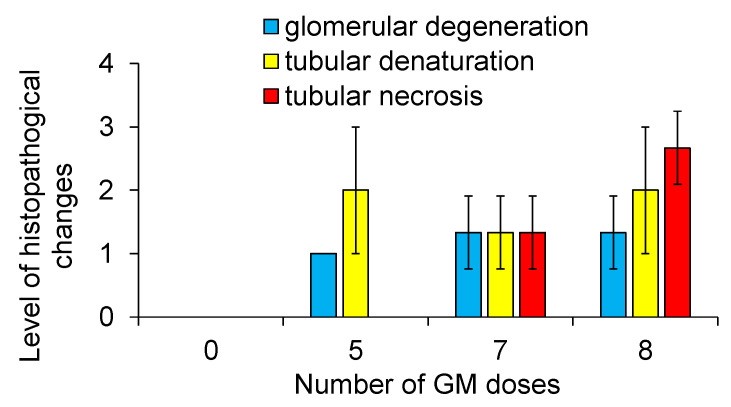
Histopathological changes in the kidneys of rats treated with gentamicin (GM). GM was intraperitoneally injected at single doses of 0.25 g/kg body weight per administration at 24 h intervals. Data were obtained from microscopic histological analysis of kidney sections by visual inspection. Pathological cases of absent (0), mild level (1), moderate level (2), and severe level (3) changes compared to the normal kidneys of rats treated with saline. Data are expressed as mean ± SD (*n* = 3 rats).

**Figure 5 diagnostics-13-01823-f005:**
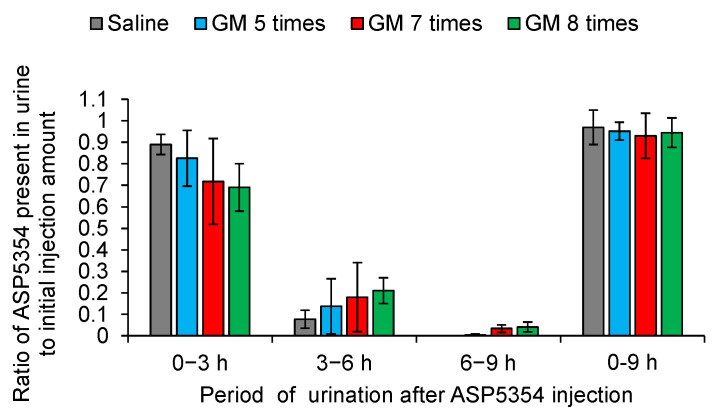
Time course of ASP5354 present in the urine excreted from the bladder after ASP5354 injection in gentamicin (GM)-treated rats. ASP5354 was intravenously injected at a single dose of 120 nmol/kg body weight in rats, administered intraperitoneally with saline (5 mL/kg body weight per administration at 24 h intervals, 5 times) or GM (0.25 g/kg body weight per injection at 24 h intervals, 5, 7, or 8 times). Data are expressed as mean ± SD (*n* = 5 rats) of the ratio to the initial injection amount of ASP5354.

**Figure 6 diagnostics-13-01823-f006:**
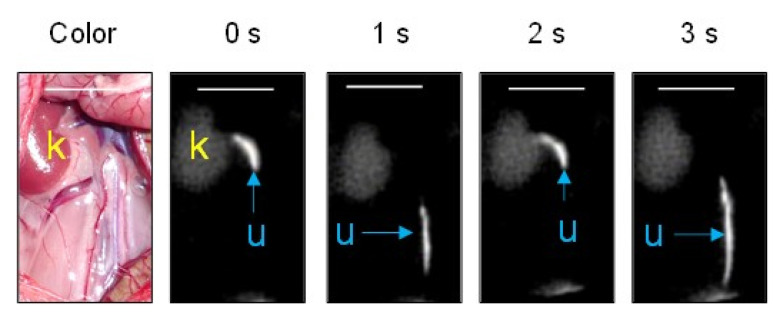
Representative near-infrared fluorescence (NIRF) images of the abdominal cavity of a control rat after intravenous injection of ASP5354. NIRF was measured at an excitation of 1.0 a.u., 5 min after intravenous injection of ASP5354 at a single dose of 120 nmol/kg body weight in rats treated with an intraperitoneal injection of saline (5 mL/kg body weight per administration at 24 h intervals, 5 times). NIRF images for 3 s are fragmentarily demonstrated. NIRF is displayed in white. k, kidney; u, ureter. Scale bar, 10 mm.

**Figure 7 diagnostics-13-01823-f007:**
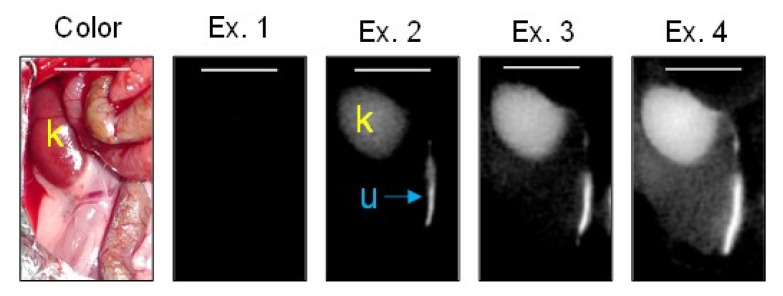
Representative near-infrared fluorescence (NIRF) images of the abdominal cavity of a control rat after intravenous injection of ASP5354. NIRF was measured at excitations of (Ex.) 1.0, 2.0, 3.0, and 4.0 a.u., 5 min after ASP5354 injection at a single dose of 40 nmol/kg body weight in rats treated with intraperitoneal injection of saline (5 mL/kg body weight per administration at 24 h intervals, 5 times). NIRF is displayed in white. k, kidney; u, ureter. Scale bar, 10 mm.

**Figure 8 diagnostics-13-01823-f008:**
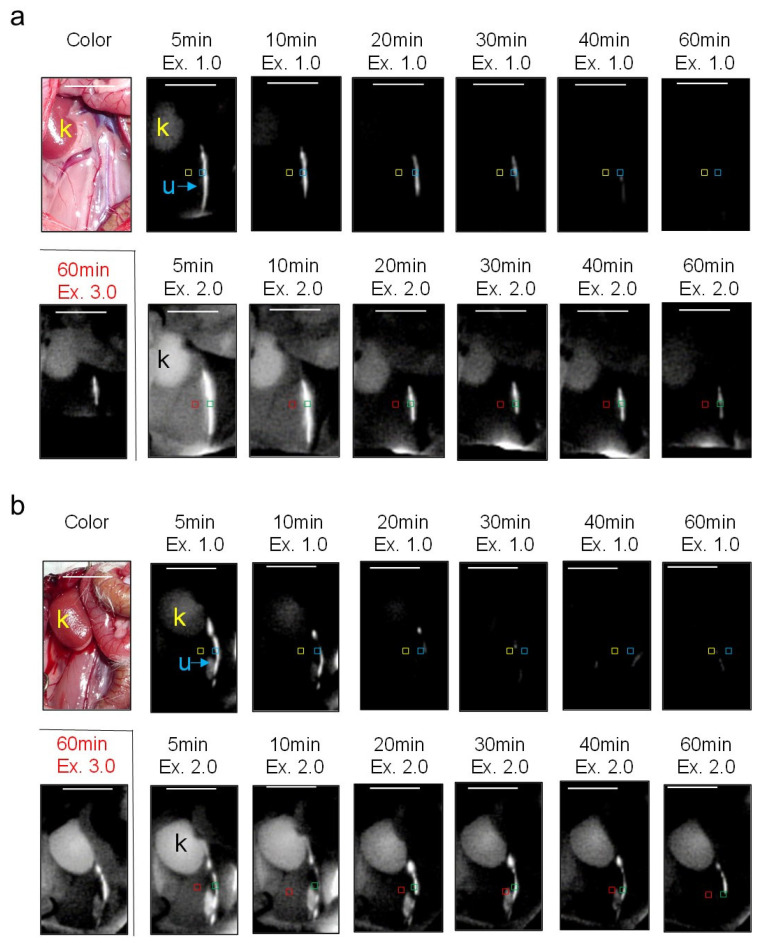
Representative near-infrared fluorescence (NIRF) images of rat ureters using ASP5354. NIRF was measured after intravenous injection of ASP5354 at a single dose of 120 nmol/kg body weight in rats treated with saline (5 mL/kg body weight per administration at 24 h intervals, 5 times) (**a**), gentamicin at 0.25 g/kg body weight per injection at 24 h intervals 5 (**b**), 7 (**c**), or 8 (**d**) times. NIRF was detected at an excitation of Ex. 1.0, 2.0, and 3.0 a.u. Blue, green, yellow, and red squares represent the ROIs to evaluate NIRF intensities in ureters at Ex 1.0 and 2.0, tissues around the ureters at Ex. 1.0, and tissues around the ureters at Ex. 2.0 a.u., respectively. Bladders were covered because NIRF in the bladders was too strong, and interfered with the NIRF of the surrounding tissues. NIRF is displayed in white. k, kidney; u, ureter. Scale bar, 10 mm. (*n* = 3 rats).

**Figure 9 diagnostics-13-01823-f009:**
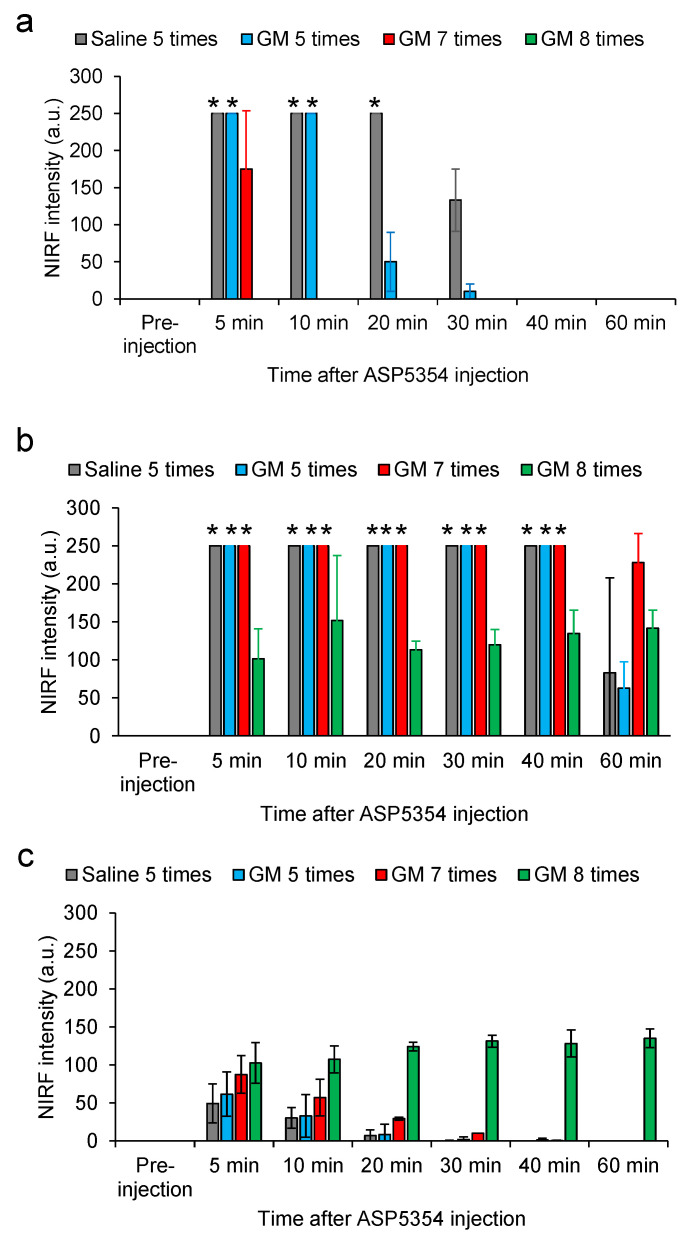
Near-infrared fluorescence (NIRF) intensity in rat tissues after ASP5354 intravenous injection. (**a**,**b**) Ureter NIRF intensities were evaluated where outlined in blue and green squares in Figure 8 (images in Figure 8 are typical examples), for Ex. 1.0 and Ex. 2.0 a.u., respectively. The evaluation points were set halfway between the kidney and bladder. * NIRF intensity exceeded the maximum detection limit. (**c**) NIRF intensity in tissues around the ureters was evaluated within the red squares depicted in Figure 8 (Ex. 2.0 a.u.). Data are expressed as mean ± SD (*n* = 3 rats).

**Figure 10 diagnostics-13-01823-f010:**
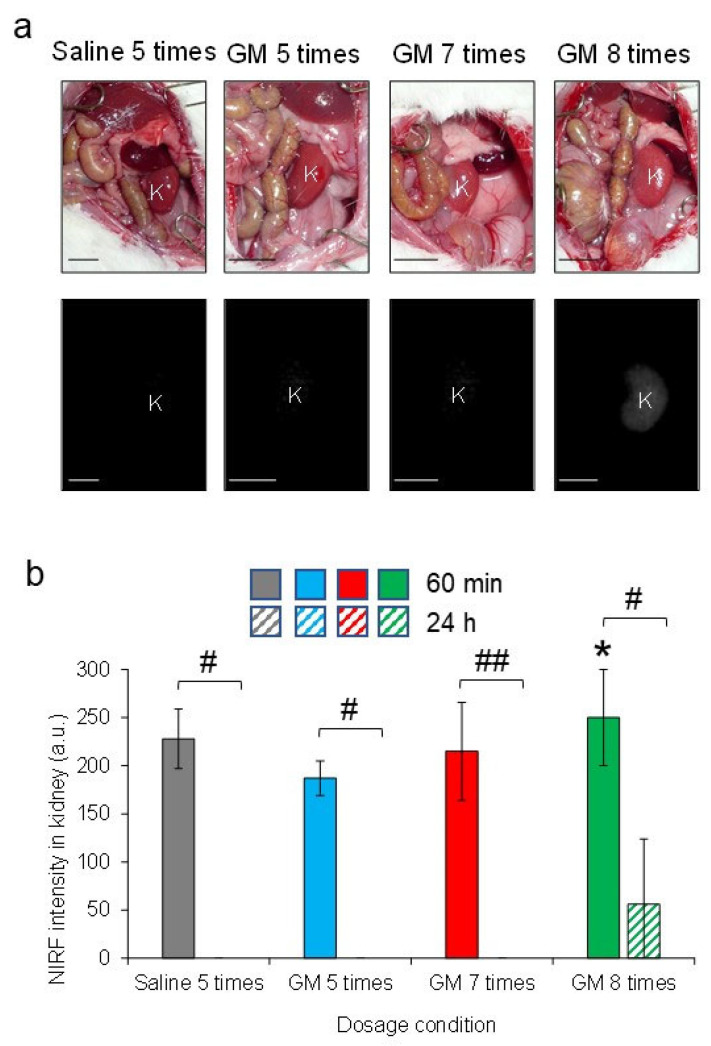
Near-infrared fluorescence (NIRF) imaging of rat abdominal cavity after intravenous injection of ASP5354. (**a**) Representative NIRF images at excitation level 3.0 a.u. NIRF was measured 24 h after intravenous injection of ASP5354 at a single dose of 120 nmol/kg body weight in rats 24 h after treatment with saline (5 mL/kg body weight per intraperitoneal administration at 24 h intervals, 5 times) or gentamicin (GM) at single doses (5, 7, or 8 times) of 0.25 g/kg body weight per intraperitoneal administration at 24 h intervals. NIRF is displayed in white. k, kidney. Scale bar, 10 mm. (**b**) NIRF intensity at excitation level 3.0 a.u. in the kidney 60 min and 24 h after the injection of ASP5354. The NIRF measurement method is described in (**a**). Data are expressed as mean ± SD (*n* = 3 rats). * NIRF intensity exceeded the maximum detection limit. # *p* < 0.0005 and ## *p* < 0.005 indicate significant difference in each pair.

**Figure 11 diagnostics-13-01823-f011:**
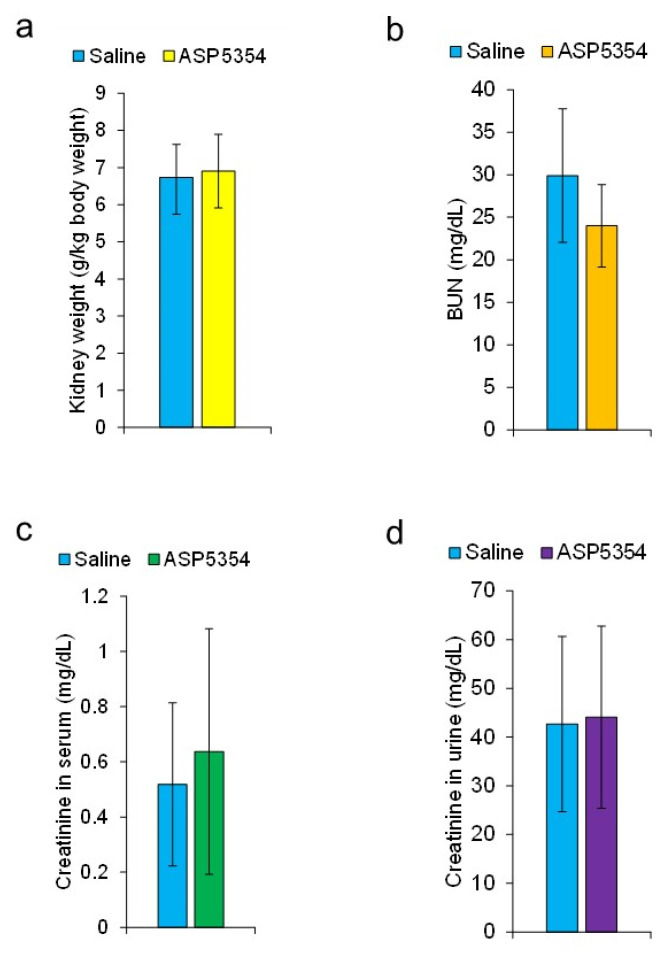
Effect of ASP5354 injection on characteristic parameters of gentamicin (GM)-induced nephrotoxicity. Saline (5 mL/kg body weight) or ASP5354 (a single dose of 120 nmol/kg body weight) was intravenously injected in rats 24 h after intraperitoneal administration of GM at single doses of 0.25 g/kg body weight per administration at 24 h intervals 7 times. The parameters were measured 6 days after the saline or ASP5354 injection. (**a**) Kidney weight. (**b**) Blood urea nitrogen (BUN). (**c**) Creatinine in serum. (**d**) Creatinine in urine. Data are expressed as mean ± SD (each experiment *n* = 5 rats).

**Figure 12 diagnostics-13-01823-f012:**
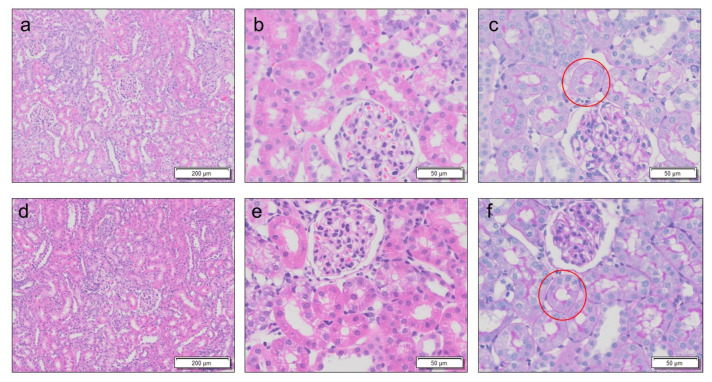
Representative photomicrographs of kidneys 6 days after treatment with ASP5354 or saline in rats with moderate AKI. Saline (5 mL/kg body weight) or ASP5354 (a single dose of 120 nmol/kg body weight) was intravenously injected in rats 24 h after intraperitoneal administration of gentamicin (GM) at single doses of 0.25 g/kg body weight per intraperitoneal administration at 24 h intervals 7 times. The kidneys were removed 6 days after the saline or ASP5354 injection. (**a**–**c**) Saline injection. (**d**–**f**) ASP5354 injection. The photomicrographs (**a**,**b**,**d**,**e**) illustrate hematoxylin and eosin-stained tissues. The photomicrographs **c** and **f** illustrate periodic acid-Schiff-stained tissues. Red circles present representative regeneration of proximal tubules.

## Data Availability

Data is contained within the article.

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
