# Peer review of "Evaluation of the Utilization of Near-Infrared Fluorescent Contrast Agent ASP5354 for In Vivo Ureteral Identification in Renal Diseases Using Rat Models of Gentamicin-Induced Acute Kidney Injury"

_diagnostics, 2023, doi:10.3390/diagnostics13101823_

Round 1

Reviewer 1 Report

In this manuscript, the authors employed a near-infrared fluorescence contrast agent ASP5354 for ureteral identification with acute kidney injury (AKI). Mild, moderate, and severe AKI model induced by gentamicin were successfully established, and confirmed by serum renal index detection and histological staining. After the intravenous injection of ASP5354 into rats with mild and moderate AKI, the ureters were clearly imaged at a high ratio of NIRF intensity in the ureter/NIRF intensity in the tissues around the ureter. ASP5354 in rats with moderate AKI did not affect the biochemical kidney functions or histopathological conditions of kidney tissues compared to no injection of ASP5354. It is an intersting work that can be considered for publication in Diagnostics after revisions.

1. Although the authors have synthesized the near infrared fluorescence imaging molecule ASP5354, its structure needs to be shown in this manuscript for the convenience of readers' understanding.

2. Authors intensively use abbreviations. It would be beneficial to work to add a list of them.

3. Recent studies about the imaging of acute kidney injury are suggested to refer (Nature Materials, 2019, 18: 1133–1143; Small, 2021, 17: 2005113; Biomaterials, 2021, 271: 120706).

Author Response

Response to Review

Thanks for your comments. I carefully considered the comments from the reviewer 1.

Reviewer: 1

Comment 1: Although the authors have synthesized the near infrared fluorescence imaging molecule ASP5354, its structure needs to be shown in this manuscript for the convenience of readers' understanding.

Response 1: The chemical structure is shown as new Figure 1.

Comment 2: Authors intensively use abbreviations. It would be beneficial to work to add a list of them.

Response 2: Abbreviations section is added above References section.

Comment 3: Recent studies about the imaging of acute kidney injury are suggested to refer (Nature Materials, 2019, 18: 1133–1143; Small, 2021, 17: 2005113; Biomaterials, 2021, 271: 120706).

Response 3: These and other new references are added as references 12–16, 27, and 28.

Reviewer 2 Report

In this study, the authors evaluated several NIRF detection conditions for ASP5354 to test whether it could be applied at ureteral imaging using rat models with gentamicin induced mild, moderate, and severe acute kidney injury (AKI). The authors also investigated whether ASP5354 has potential nephrotoxicity. They found that ASP5354 could be a potential contrast agent applying for intraoperative ureteral identification in patients with renal injury. This study provides new knowledge for a novel agent that may have potential to be applied in clinical purpose. Some comments below may help to improved the quality of this article. 

1. Are there typing errors of the p value p<0.0005, p<0.1 and p>0.1 in line 216 and line 217? These p values are not commonly used.

2. The administration of ASP5354 was via intravenous injection and it could distribute to whole body. Will ASP5354 accumulate in other organs in AKI or CKD rats or patients? How to prove that all ASP5354 can be excreted into urine? The toxicity information of other organs should be discussed. 

Some grammatical errors in the text and figure legends need to be checked carefully and revised. Writing style could be improved to make the article more clear and easier to be understood.

Author Response

Response to Review

Thanks for your comments. I carefully considered the comments from the reviewers.

Reviewer: 2

Comment 1: Are there typing errors of the p value p<0.0005, p<0.1 and p>0.1 in line 216 and line 217? These p values are not commonly used.

Response 1: These values are deleted.

Comment 2: The administration of ASP5354 was via intravenous injection and it could distribute to whole body. Will ASP5354 accumulate in other organs in AKI or CKD rats or patients? How to prove that all ASP5354 can be excreted into urine? The toxicity information of other organs should be discussed.

Response 2: As shown in Figure 5, ASP5354 of 95, 93, and 94% of ASP5354 injected in mild, moderate, and sever, respectively, AKI rats was excreted in 9 h after ASP5354 injection. However, we had no data showing whether all ASP5354 was excreted in the urine, showing which organs contained trace amounts of ASP5354, and showing whether residual ASP5354 affected the body. We discuss based on this information in Lines 366-377 in Discussion section as limitations in this study.

Comment 3: Comments on the Quality of English Language

Some grammatical errors in the text and figure legends need to be checked carefully and revised. Writing style could be improved to make the article more clear and easier to be understood.

Response 3: This manuscript was checked by Editage for the English language editing and finally by us. The service provided a confirmation certificate. We submitted the certificate to MDPI office by email and attach in next page. Moreover, we recheck the manuscript and correct inappropriate expressions for resubmission.
